# Ex Vivo Optical Coherence Tomography Analysis of Resected Human Bladder with a Forward-Looking Microelectromechanical Systems Mirror-Based Catheter

**DOI:** 10.3390/s25185794

**Published:** 2025-09-17

**Authors:** Marinka J. Remmelink, Paul R. Bloemen, Patrick van der Voorn, Xavier Attendu, Richard M. van den Elzen, Jakko A. Nieuwenhuijzen, Jorg R. Oddens, Ton G. van Leeuwen, Daniel M. de Bruin

**Affiliations:** 1Department of Urology, Amsterdam UMC Location University of Amsterdam, Meibergdreef 9, 1105 AZ Amsterdam, The Netherlands; 2Cancer Center Amsterdam, Cancer Treatment and Quality of Life, Boelelaan 1118, 1081 HV Amsterdam, The Netherlands; 3Department of Biomedical Engineering and Physics, Amsterdam UMC Location University of Amsterdam, Meibergdreef 9, 1105 AZ Amsterdam, The Netherlands; 4Department of Pathology, Amsterdam UMC Location Vrije Universiteit Amsterdam, De Boelelaan 1117, 1081 HV Amsterdam, The Netherlands; 5Department of Urology, Amsterdam UMC Location Vrije Universiteit Amsterdam, Boelelaan 1117, 1081 HV Amsterdam, The Netherlands; 6Cancer Center Amsterdam, Imaging and Biomarkers, Boelelaan 1118, 1081 HV Amsterdam, The Netherlands

**Keywords:** bladder cancer, optical coherence tomography, optical imaging, microelectromechanical systems

## Abstract

A technique that enables real-time diagnosis of bladder cancer is needed. Optical coherence tomography (OCT) is a promising technique, but a forward-looking OCT catheter is necessary for OCT to enable bladder cancer diagnosis. This study aims to describe the design of a novel forward-looking microelectromechanical systems (MEMS)-based OCT catheter, assess the performance characteristics, and evaluate its ability to identify histopathological characteristics of bladder specimens. A description of the OCT catheter and systems used is provided. Performance characteristics were measured with a beam profiler and microscopy slide (mirror for dispersion and thickness for lateral calibration). Ex vivo measurements were performed on resected bladder tissue from patients undergoing a radical cystectomy. A forward-looking OCT probe with an outer diameter of 2.52 mm and a rigid length of 17 mm was designed and evaluated. The focus position was measured as 10.9 mm from the MEMS mirror, with a Rayleigh length of 2.55 mm. Several histopathological features could be correlated to OCT features of the ex vivo measurements. In conclusion, a forward-looking OCT probe that can be inserted in the working channel of a rigid cystoscope was designed and evaluated. Performance characteristics were overall in line with simulated expectations.

## 1. Introduction

Bladder cancer (BC) continues to pose a significant healthcare challenge worldwide, with a rising number of new diagnoses, reaching 614,298 in 2022 [1]. Currently, the diagnosis of BC relies on initial cystoscopic evaluation at the outpatient clinic and histopathology obtained during a subsequent transurethral resection of the bladder tumor (TURBT) [2]. With recurrences in >50% of patients, many patients undergo this diagnostic process multiple times [3]. Therefore, this diagnostic process imposes a considerable burden on patients and results in substantial healthcare costs [4,5]. The reliance on histopathology comes with inter-observer variability and inevitable delays between suspicion and diagnosis of BC, which impact suitable treatment choice and the moment of initiation [6].

In an ideal scenario, diagnosing BC would take place during first outpatient cystoscopic evaluations. With a real-time histopathological evaluation of the tissue in vivo, the invasion and the aggressiveness of the (cancer) cells could be determined by visualizing and quantifying the morphological tissue differences. This approach would accelerate the diagnostic process, facilitate prompt initiation of appropriate treatments, and potentially avoid unnecessary surgeries [7]. Additionally, it would enable the safe application of other treatment options in selected patients, such as laser fulguration and active surveillance, when the BC stage and grade were known [8,9,10].

Several methods to obtain an ‘optical biopsy’ of BC, including confocal laser endomicroscopy (CLE) [11,12,13], endocytoscopy [14], optical coherence tomography (OCT) [15,16,17], and Raman spectroscopy, have been described [18,19,20]. Among these, OCT has shown promise, particularly due to the improvements in its field over recent decades [21,22]. OCT utilizes light, in most cases infrared light, to create high-resolution depth-resolved cross-sectional images of tissue structures by measuring the interactions of light. Most available OCT catheters are side-viewing and have a focus within 1 cm from the tip [23,24]. Due to geometric mismatches, these probes are not suitable for imaging the interior of spherical shapes. To overcome this limitation, we have developed a forward-looking OCT catheter that utilizes a combination of a microelectromechanical systems (MEMS)-driven scanning mirror and a gradient index micro lens (GRIN). Because the human bladder is typically filled during cystoscopy with saline solution in water, this imaging catheter has been designed to operate with a 1064 nm OCT system to ensure minimal water absorption while still maintaining adequate imaging depth [25,26].

The aims of this study are to describe the design of this forward-looking OCT catheter, assess its performance characteristics, and evaluate its ability to identify tumor tissue and other histopathological characteristics of the bladder wall ex vivo.

## 2. Materials and Methods

### 2.1. OCT System and Catheter Description

#### 2.1.1. OCT System Set-Up

The ex vivo OCT set-up was composed of three parts. First, there was the newly designed MEMS mirror-based OCT catheter, which was mounted on a motorized X-Y-Z platform to allow precise movement of the bladder specimen beneath the catheter. Second, there was the HSL 1 High-speed Vertical-Cavity Surface-Emitting Laser (VCSEL) swept-source laser (Santec, Fukuoka, Japan) that operated at up to 400 kHz. The 400 kHz swept-source OCT module was chosen in order to have enough A-scans per B-scan given the MEMS scan frequency in the OCT catheter, with minimum roll-off. The central wavelength of the HSL 1 system was 1054 nm with an optical bandwidth of 70 nm, resulting in a simulated axial resolution of 7 µm in air. The output power of the VCSEL was 26 mW, with a resulting output power of 6 mW at the catheter tip. The third part was the IVS-1000 interferometer (Santec, Fukuoka, Japan), with a customized reference arm length, and a data acquisition system consisting of an HAD-5200-B DAQ card (Santec, Fukuoka, Japan). Custom-designed electronics and software were used to synchronize the data acquisition with the MEMS mirror. A guide laser was incorporated to provide real-time information on which area was scanned. A schematic and a real-life depiction of the set-up are provided in Figure 1.

#### 2.1.2. Design Properties of the MEMS-Based OCT Catheter

An optomechanical catheter was designed for eventual in vivo imaging, housing both focusing optics and an MEMS mirror (Scinvivo BV, Eindhoven, The Netherlands, patent number: NL2020564B1). The outer diameter of the catheter was 2.52 mm, with a rigid length of 17 mm, making it suitable for the working channel of a rigid cystoscope. The alignment of the optics in relation to the MEMS mirror was achieved with top and bottom support, allowing precise positioning of the fiber and optics with respect to the MEMS mirror. The optics consisted of a GRIN lens (Grintech GmbH, Jena, Germany) with a prism attached to create a simulated focus in air at 11 mm from the catheter tip. The GRIN lens–prism combination was glued to the single-mode fiber (SMF-28). A detailed schematic representation of the OCT imaging catheter is shown in Figure 2.

The converging light from the GRIN lens was reflected by the MEMS mirror towards the tissue sample. The MEMS mirror was controlled by an application-specific integrated circuit (ASIC), which was located at the tip of the catheter just behind the MEMS mirror (see Figure 2). The MEMS mirror consisted of a moving and a static part. The moving part of the MEMS, the mirror, was held in place by torsion bars. These bars were connected to the static part of the MEMS, which was again attached to the bottom support. The torsion beams enabled movement of the mirror, creating the scanning motion of the OCT beam. The ASIC driver supplied a square wave signal of 15 V to the static part of the MEMS mirror, resulting in an oscillating motion of the moving part of the MEMS mirror. The frequency at which the MEMS mirror oscillated was 550–565 Hz, which was defined by the mass of the MEMS mirror, as well as the stiffness of the supporting torsion bars that held the mirror in place within the catheter. The ASIC driver further comprised a feedback input to receive a feedback signal indicative for a rotational state of the rotor vs. the stator of the MEMS mirror. Consequently, single B-scans consisted of ~364 A-scans, when the MEMS mirror was paired with a swept source laser producing A-scans at a rate of 200 kHz. To reduce the lateral step size, two successive B-scans were consolidated into a single image frame with ~728 A-scans. It was required that the corresponding A-scans in the successive B-scans were slightly offset with respect to each other. The final frames per second (FPS) rate was approximately 69 Hz after averaging four combined B-scans (with each consisting of ~728 A-scans).

### 2.2. OCT System and Catheter Performance Analysis and Evaluation

The OCT measurements needed for OCT system and catheter performance analysis were performed in between the OCT measurements on resected bladder specimens. The simulated performance values (focus position and Rayleigh length) were modeled with Ansys Zemax Optic Studio (Ansys Inc., Canonsburg, PA, USA) simulation, based on a scan angle of 30 degrees and a lateral scanning range of 5 mm.

#### 2.2.1. Waist and Rayleigh Length Determination

The real performance of the MEMS-based OCT imaging catheter, in combination with the OCT system, was evaluated in multiple steps. The lateral and axial pixel sampling were 716 and 1024, respectively. The lateral resolution, focus position, and Rayleigh length of the system were measured with a scanning slit beam profiler (BP209-IR, Thorlabs GmbH, Newton, NJ, USA), which was positioned on a linear translation stage. By translating the beam profiler through the focus, both the divergence and the lateral resolution were measured. The beam waist functions in 2 dimensions (X and Y) of the OCT beam were calculated for each dimension with the following function:$$\omega_{z}=\omega_{0}\sqrt{1+{\left(\frac{{Z-Z}_{f}}{Z_{r}}\right)}^{2}}$$
in which ω_0_ is the radius of the beam at focus (µm), *Z_r_* is the Rayleigh length (µm) and *Z* is the position in depth, and *Z_f_* is the position of the focus (µm).

At different positions in depth, OCT measurements of a reflector (1 mm thick microscopy slide) were performed. These measurements were used to determine the axial resolution and calculate the dispersion compensation. The dispersion compensation was performed according to the method of Attendu et al. [27].

#### 2.2.2. Cartesian Pixel Remapping

The scan angle and distance from the MEMS mirror to 0-delay were needed for correctly displaying the OCT image in a cartesian plot. The uncorrected merged OCT images of the edge of the microscopy slide, at different axial positions, are depicted in Figure 3B. The microscopy slide contained bevels on both sides. The lateral edge points of the microscopy slide bevels were marked.

For different combinations of the scan angle and distance from the MEMS mirror to 0-delay, the OCT images were remapped into cartesian plots. For these combinations, the difference between each microscopy slide width in the resulting cartesian plot with a known width (1 mm) was calculated. The sum of differences in the projected and known width of the microscopy slide at different axial positions was calculated. Example: D_1_ + D_2_ + D_3_ + etc., where D_1_ is the difference between the width of the projected microscopy slide and the actual thickness (1 mm) at location 1, D2 is this difference at location 2, etc. These sums were calculated for all combinations of angles between 7 and 8.5 degrees and distances to 0-delay from 7 to 8 mm, with steps of 0.01 degrees and mm, and were plotted in a false color plot (see Figure 3A). The optimal combination of the angle and distance to 0-delay was the sum closest to zero; this point is indicated with a red marker in Figure 3A.

### 2.3. Ex Vivo Measurements on Bladder Specimens

#### 2.3.1. Study Design

This study was approved by the Institutional Review Board (2022.0699) of the Amsterdam University Medical Center. Written informed consent was obtained from all patients before enrolment. The study was performed in accordance with the Guidelines for Good Clinical Practice (ICH E6 (R1)).

#### 2.3.2. Study Population

Patients were prospectively included at the Amsterdam University Medical Center (Amsterdam, The Netherlands). Adult patients who were scheduled to undergo a radical cystectomy for bladder cancer were eligible for participation. Exclusion criteria included neo-adjuvant chemotherapy without residual tumors on imaging, previous radiotherapy of the bladder, and only carcinoma in situ present. These patients were excluded because we aimed to visualize evident bladder tumors and because the bladder structure could be different due to radiotherapy treatment. The anticipated number of patients to be included and have measurements performed on bladder specimens was up to 10.

#### 2.3.3. Tissue Handling

Directly after surgical whole bladder removal, a fresh 2 by 2 cm tissue sample with suspected tumor and normal bladder tissue was selected by the pathologist and researcher, cut out, and pinned on a cork plate (see Figure 1B). Directly thereafter, the OCT measurements were performed while keeping the samples covered with saline to prevent tissue dehydration. The number of B-scans captured during the measurements depended on the size of the bladder sample. After the measurements, the direction of the B-scans was marked on the cork plate with ink. When the OCT measurements were finished, the bladder specimen was fixed in formalin and returned to the Department of Pathology for regular histopathological processing (paraffin embedding, cutting slides, and hematoxylin and eosin staining).

During this specimen handling, the slices were cut accordingly to the B-scan direction, indicated by the researcher, in order to create a histological image comparable to the OCT image. The histopathological diagnosis of the tissue sample was defined by the pathologist, who was blinded to the OCT results. Histopathology results were viewed and discussed by the pathologist (PvdV) and researcher (MJR). Tumor presence and locations were discussed, as well as other notable histopathological features.

#### 2.3.4. OCT Image Analysis

Multiple B-scans were acquired adjacent to each other with a known spacing between each scan (see Figure 4). Matlab version 9.11.0—R2021b (The MathWorks Inc., Natick, MA, USA) was used to remove fixed pattern noise and to stitch the OCT B-scans to form ‘slides’. To improve continuity of structures within the OCT images, minor adjustments in the angle of the B-scans were manually made based on visual judgment of the resulting stitched OCT image.

OCT images were compared to the histopathological slides based on their location within the bladder specimen and on visual characteristics/landmarks, such as the contour, and recognizable histopathological characteristics, such as cysts.

## 3. Results

### 3.1. OCT System and Catheter Performance

Simulated characteristics of catheter optics from the Zemax simulation resulted in a simulated Rayleigh length of 2.0 mm in air (2.64 mm in water) and a focus point at 7.7 mm from the MEMS mirror in air (10.3 mm in water) (see Table 1).

The results of the beam profiler measurements revealed the focus position at 3.4 mm from zero delay with a Rayleigh length of 2.55 mm and a beam waist in focus of 28.6 µm. The measured and calculated spot size of the beam in depth, measured by the beam profiler, are presented in Figure 5.

The beam waist in 2 dimensions (X and Y) of the optical coherence tomography (OCT) beam were calculated for each dimension with the function $\omega_{z}=\omega_{0}\sqrt{1+{\left(\frac{{Z-Z}_{f}}{Z_{r}}\right)}^{2}}$ in which ω_0_ is the radius of the beam at focus (µm); *Z_r_* is the Rayleigh length (µm); *Z* is the position in depth; and *Z_f_* is the focus position (µm).

The axial resolution was 11 µm in air (8 µm in tissue) after dispersion compensation. The axial scan range was 8.22 mm. The calculated optimal angle and distance to 0-delay were 7.84 degrees and 7.46 mm, respectively (see Figure 3). Then, the resulting distance from the MEMS mirror to the focus point was 10.9 mm. The dimensions of the OCT B-scan are visualized in Figure 6, and the experimental characteristics of the system and catheter are listed in Table 1. Furthermore, the laser source exhibited a coherence length exceeding 100 m [28]; therefore, system roll-off was deemed negligible.

### 3.2. Ex Vivo Measurements Results

#### 3.2.1. Study Population

Between December 2022 and March 2024, ten patients were included in the ex vivo experiment. The bladder specimens of three patients could not be measured with OCT, due to system failure prior to the measurements. A total of seven bladder samples were measured with the OCT system. The OCT measurements of the last specimen were performed after a complete and adequate system performance analysis, including dispersion calibration. Inclusion stopped prematurely after seven completed measurements, due to probe breakdown.

#### 3.2.2. OCT Image Comparison to Histopathology

The OCT ‘slides’ of these first three measurements showed gaps in between the OCT images (see Figure 7A,C,E), which was in retrospect due to a smaller angle of the B-scan than anticipated. During the following measurements, the B-scan angle and distance between B-scans in two perpendicular directions (see Figure 4) were noted and filed, along with a schematic drawing of the locations of the measurements within the bladder specimen, to improve the stitching and comparison of the OCT images (see Figure 7G,I,K).

Histopathology showed tumors in all samples, all invading the musculus detrusor. In most samples, atrophic urothelium was observed. In one sample, carcinoma in situ was present and a superficial tumor invading the submucosa was seen.

Several structures are seen on the OCT images and can be correlated to the histopathology. These include a part of the ureter crossing the sample, a Von Brunn nest (non-neoplastic reactive urothelial lesion) (Figure 7E,F), a small tumor necrotic cavity, shown in the co-localized hematoxylin and eosin-stained histopathology and OCT images as an empty space due to local phagocytosis (Figure 7I,J) [29], the effect of chemotherapy (fibrosis) (Figure 7K,L), and irregularities in the contour of the sample (Figure 7C–F,I,J). In one sample, higher intensity was observed at the location of the tumor in the correlating histopathology.

In the OCT images, no evident layers could be distinguished. In Figure 7, representative OCT images of five patients are shown with their corresponding hematoxylin and eosin-stained histopathology images and marked correlations.

In some OCT images, when scrolling through, possible vascular structures are observed (see Appendix A), although a direct correlation to the histopathology could not be made.

## 4. Discussion

This is the first ex vivo study with the novel MEMS-based forward-looking OCT catheter interfaced with a 200 kHz OCT system. System performance analysis demonstrated a focus position at 10.9 mm in air from the MEMS mirror with a Rayleigh length of 2.55 mm, an axial resolution of 8 µm, and an FPS of 69 Hz, aligning relatively well with the initial design goals. However, the B-scan angle appeared to vary over time, and the focus distance differed minimally from the simulated values. Several structures were identifiable and could be correlated with histopathological observations. With visual analysis, no distinct layered architecture was distinguishable in the OCT images. The ability to differentiate these layers of the bladder wall is needed to identify the depth of invasion of tumors.

To assess a spherical organ like the bladder with OCT, a forward-looking OCT catheter is necessary. Side-viewing helical scanning OCT systems are unsuitable for bladder wall imaging, as it is difficult, and for some regions in the bladder even impossible, to position a catheter parallel to the bladder wall. To our knowledge, three other forward-looking OCT catheters have been developed to date, namely, the Niris Imalux and two custom-made catheters [17,30,31]. However, the Niris Imalux is no longer available on the market [30], one custom-made catheter has not been commercially released [31], and the other custom-made catheter has information only in Russian, decreasing its findability and usability, and has no CE certification [17]. The focus distance and field of view (FOV) of Ren et al.’s custom-made catheter is comparable to those of the catheter described in this paper. However, the Niris Imalux and the custom-made catheter OCT1300-U required direct tissue contact, limiting its FOV to the size of the catheter tip [17,30]. After dispersion compensation, the new OCT catheter presented in this study offers slightly improved axial resolution (~8 µm) compared to Ren et al.’s (~10 µm), OCT1300-U (~20 µm), and the Niris Imalux (10–20 µm). The axial resolution is not related to the new catheter design but is determined by sweeping with the OCT system. In contrast, all other OCT catheters have better lateral resolution, with the Niris Imalux having a lateral resolution of ~20–25 µm, the OCT1300-U having one of ~25 µm, and Ren et al.’s catheter having one of ~12 µm [17,30,31]. However, the larger focal spot size (worse lateral resolution) increases the Rayleigh length, which is not necessary in contact probes like the Niris Imalux or the OCT1300-U but improves tissue visualization in non-contact probes.

In comparison with more widely studied side-viewing helical scanning OCT systems and catheters, such as the Dragonfly Optis and the catheter developed by Li et al., the OCT system and new catheter described here also exhibits better axial resolution (8 µm compared to ~15 µm) [23,24]. The lateral resolution of Li et al.’s catheter (9.6 µm) surpasses that of the new catheter, while the Dragonfly Optis shows similar lateral resolution (20–40 µm). Both side-viewing catheters have a focus closer to the tip (3 mm and 0.47 mm) and shorter Rayleigh lengths (0.18 mm and 0.65 mm), making it more challenging to maintain the area of interest within the focal range. Although the side-viewing catheters provide a larger FOV (360 degrees), they are less practical for OCT measurements inside the bladder compared to the forward-looking design of the new catheter.

A notable finding of the performance analysis is that the angle of the B-scan differs significantly from the expected angle. During the measurements, it was noticed that the angle of the B-scan, and thereby the optimal MEMS frequency, varied slightly over time. This variation arises from the inherent dependence of the scan frequency on the MEMS mirror’s resonance frequency—which is determined by the mirror’s mass and the stiffness of the torsion beams—and dependence of the scan angle on the applied driving voltage. The MEMS mirror, fabricated using standard silicon-based semiconductor processes, currently cannot maintain operation at its maximum tilt angle over extended periods due to limitations in material and structural endurance. Driving the voltage to the limit, which would result in a scan angle of 30°, would result in a relatively fast degradation of the torsion bars, eventually resulting in irreversible damage to the MEMS mirror. Therefore, in the current study, where a single catheter was reused for multiple samples, the driving voltage was limited to avoid rapid degradation of the torsion beams and prevent permanent damage to the MEMS device. Despite this precaution, some degradation was still observed, resulting in a gradual drift in beam stiffness and, consequently, in the resonance frequency. In contrast, this effect is considered negligible in a further in vivo study, as the probe is designed to be single-use. Furthermore, a variation in MEMS frequency is also partially due to temperature changes, caused by use of the catheter and by the change in room temperature. During later measurements, the B-scan angle was optimized to around 20 degrees by resetting the MEMS frequency.

Another finding of the performance analysis is a discrepancy between the simulated focus distance (7.7 mm) and the experimental focus distance (10.9 mm). This can be explained by small differences in the real optical system compared to the simulated one. Through zemax simulations, it was found that a 50 micron increase in the length of the GRIN component could explain the observed shift. Such a shift would also match the observed increase in spot size and corresponding increase in Rayleigh length. It should be noted, however, that these deviations in the experimental beam profile do not compromise imaging quality.

The studies that assessed the other forward-looking catheters on bladder tissue all show that they can distinguish the different layers of the bladder and, in most studies, the presence of tumor or inflammation [15,16,17,30,31,32,33,34]. These results are in contrast with our findings. However, all studies were in vivo studies, which could explain the visibility of the layers due to even stretching of the bladder. Another potential explanation is the disruption of visible layers by pre-treatment with a TURBT, which had been performed in this study to diagnose muscle-invasive bladder cancer. Furthermore, only Manyak et al. report that they were able to identify other histopathological characteristics such as Von Brunn nests and vessels [32].

This ex vivo study has several limitations. First, the measurements were performed with a small sample size. The small sample size was partially planned because the purpose of these ex vivo measurements was to refine the measurement process and obtain example images of bladder tissue that could be directly correlated with histopathology. However, the sample size is smaller than anticipated due to premature breakdown of the OCT catheter. As the probe is designed for in vivo single use, there will not be a durability problem in future in vivo trials and clinical application. Another limitation is that the correlation between the OCT images and histopathology is not perfect. After the measurements, the sample is fixed in formalin. The fixation and cutting processes can alter the tissue’s anatomy, complicating the correlation between the OCT images and histopathological findings. In addition to the pathological characteristics of the tissue, the manual stretching method and fixation on the cork plate may have also influenced the ability to distinguish bladder layers on OCT images and the correlation of OCT with histopathology. Typically, the bladder wall is stretched by filling the bladder, which evenly stretches the layers of the bladder wall. In this experiment, the tissue sample was stretched on a small cork plate, making it difficult to achieve uniform and appropriate stretching of the layers. The uneven and insufficient stretching of the layers may have resulted in poorer image quality compared to what could be obtained in an in vivo experiment. The poorer visibility of the layers of the bladder may also be due to the fact that patients had already undergone a TURBT treatment, which probably affected the selected bladder sample. Furthermore, the correlation of the OCT images with histopathology was complicated by the fact that most samples contained little or no normal bladder tissue. In most samples, no normal urothelial layer was observed, resulting in a lack of OCT examples of normal bladder tissue. Another limitation is that the measurements were performed in air, while the OCT system and catheter are designed for measurements in water. Since the catheter used was not yet waterproof, the ex vivo measurements were conducted in air. However, the system’s wavelength is optimized for water, where it minimizes light absorption. This may have resulted in lower-quality OCT images than would be achieved in water. Furthermore, only one catheter was used during the experiments, and the impact of multiple measurements on the catheter’s performance and the resulting image quality remains unknown. Lastly, the calibration measurements were performed between the OCT imaging sessions, leading to variations in the quality of the OCT images across different tissue samples. As a result, attenuation coefficient analysis could only be performed on a single sample.

Further research is needed to evaluate the feasibility of the new OCT catheter for in vivo use. Therefore, an in vivo trial is currently ongoing. Additionally, if the current OCT system and catheter are capable of accurately assessing BC stage and grade, further development is needed to create a smaller catheter that can fit through a work channel of a flexible cystoscope, enhancing its clinical applicability. Future research should also explore real-time automatic attenuation analysis of OCT images and the application of artificial intelligence for the automated recognition of histopathological features in OCT images.

## 5. Conclusions

In this experiment, the new forward-looking MEMS-based OCT catheter was used for the first time on ex vivo bladder tissue, giving the opportunity to adjust the measurements. The characteristics of the system and catheter were mostly in line with the simulated values. Several histopathological features could be identified in OCT images. With further development of a waterproof and sterile probe, we plan to conduct in vivo research to assess the in vivo feasibility of the catheter and to investigate the diagnostic accuracy.

## Figures and Tables

**Figure 1 sensors-25-05794-f001:**
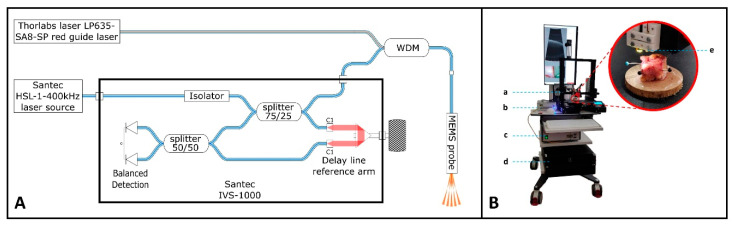
Schematic and real-life OCT system set-up. (**A**) Schematic overview of the different components within the OCT system. The HSL swept-source laser is fiber-coupled to the IVS-1000 interferometer. A portion of the light is directed towards the Wavelength Division Multiplexer (WDM), where the guide light (Thorlabs LP635) is injected into the fiber and directed towards the MEMS probe. The reflected light from the sample passes through the splitter and interferes with light from the reference arm of the set-up. (**B**) Real-life OCT system set-up consisting of a: motorized X-Y-Z platform; b: HSL 1 high-speed swept-source laser; c: IVS-1000 interferometer and data acquisition system; d: computer, and e: tip of the OCT catheter.

**Figure 2 sensors-25-05794-f002:**
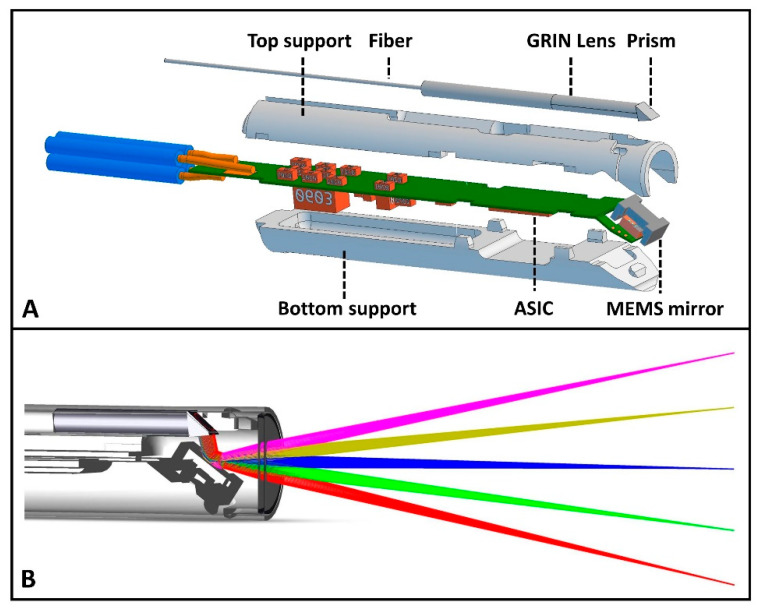
Schematic representation of the catheter design. (**A**) Schematic illustration of the components integrated into the rigid part of the catheter. (**B**) Schematic illustration of the light propagation from the prism to the MEMS mirror and out of the probe. The different light bundles illustrate the different angles at which the light is directed out of the probe.

**Figure 3 sensors-25-05794-f003:**
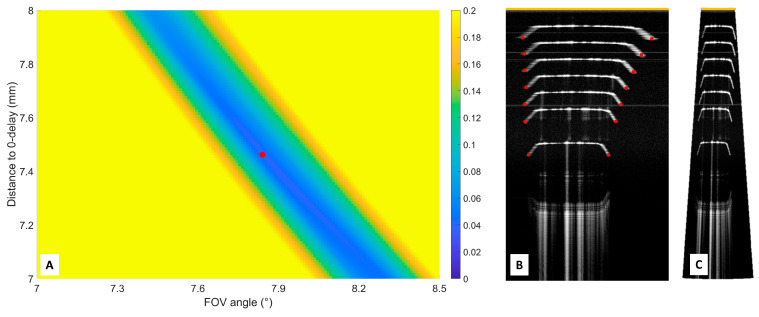
False color plot of the optimization of the scan angle and mirror-to-0-delay distance. In (**A**), the results (in mm) of the differences (compared to 1 mm) between all microscopy slide combinations of different angles and distances to 0-delay are plotted. The 0-delay point is marked with an orange bar in (**B**,**C**). The bar next to (**A**) displays the definition of the colors in the plot ranging from a summed difference of zero to 0.2 mm. In (**B**), the B-scans of the microscopy slide at different distances are merged into one B-scan, and the outer sides of the microscopy slides are marked. The microscopy slide contains bevels on both sides, so the outer side of the bevel is marked. The resulting OCT image with the optimal angle and distance to 0-delay is shown in (**C**).

**Figure 4 sensors-25-05794-f004:**
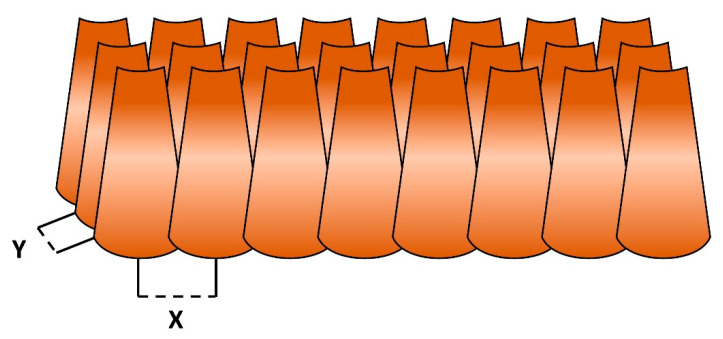
Illustrative depiction of the interrelation between individual B-scans. The independent B-scans were captured in the direction of Y with a noted spacing (Y). With the motorized X-Y-Z platform, a new row of B-scans started next to the first B-scan with a known spacing (X) in between the B-scans. The B-scans then were stitched together in the direction of X to form ‘slides’.

**Figure 5 sensors-25-05794-f005:**
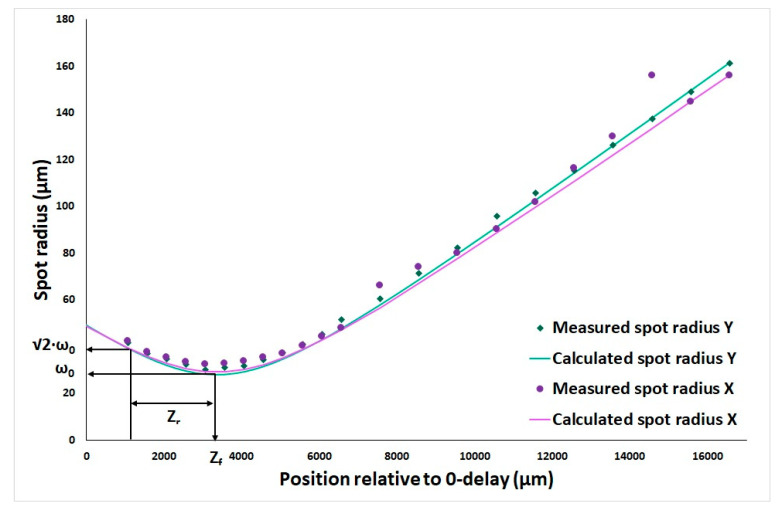
Measured spot size relative to distance from 0-delay.

**Figure 6 sensors-25-05794-f006:**
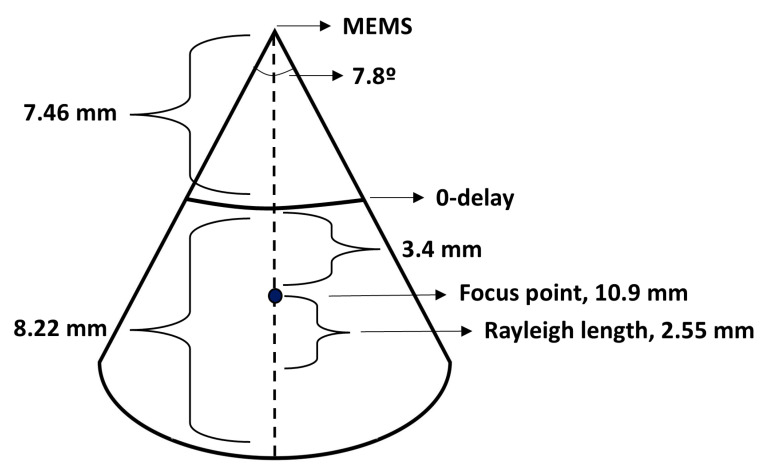
Dimensions of OCT B-scan. The dimensions in the image are not to scale. The distance from the MEMS mirror to zero delay is 7.5 mm, and the axial range of the optical coherence tomography (OCT) image is 8.2 mm. The focus point is at 10.9 mm from the microelectromechanical systems (MEMS) mirror. The Rayleigh length is 2.55 mm. With a MEMS frequency of 562 Hz, the angle is 7.8 degrees.

**Figure 7 sensors-25-05794-f007:**
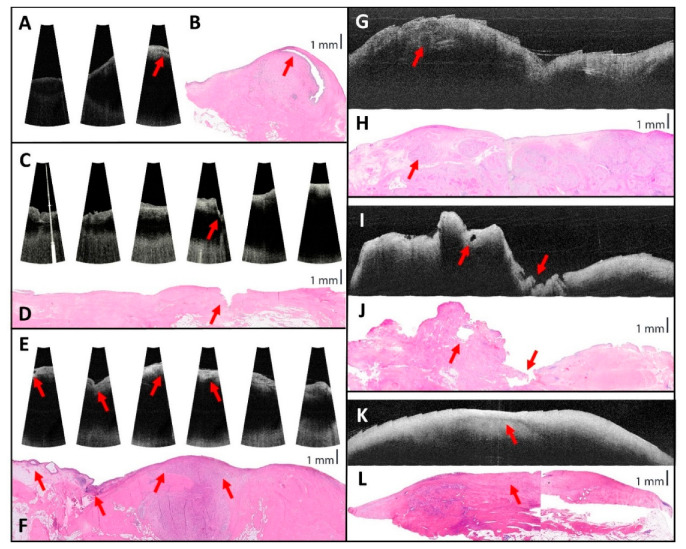
OCT images of five patients with their corresponding hematoxylin and eosin-stained histopathology slides. (**A**,**B**) Corresponding optical coherence tomography (OCT) and histopathology showing a ureter crossing the sample at the red arrow; (**C**,**D**) corresponding OCT and histopathology showing corresponding contour of the sample with a large dent at the red arrow; (**E**,**F**) corresponding OCT and histopathology showing several conformities with red arrows. From left to right: a Von Brunn nest, dent in the contour, higher-intensity OCT image at the tumor location in histopathology; the rightmost arrow also shows a higher-intensity OCT image at the tumor location in histopathology; (**G**,**H**) corresponding OCT and histopathology, showing a round tumor with the red arrow; (**I**,**J**) corresponding OCT and histopathology, showing with the left arrow a tumor necrotic cavity, appearing as a black area with high-scattering structure on top in the OCT image, and at the right deformed contour of the sample; (**K**,**L**) corresponding OCT and histopathology showing the chemotherapy effect with the red arrow, which is seen as higher intensity on the OCT image. The scalebar is applicable to the histopathology images in all directions and to the OCT images in the vertical direction.

**Table 1 sensors-25-05794-t001:** Simulated and experimental characteristics of the OCT catheter and systems.

Characteristic	Simulated Value	Experimental Value
Beam path angle	30 degrees	Varying *
Lateral scan range	5 mm	N/A
Focus point from MEMS mirror (water)	10.3 mm	N/A **
Focus point from MEMS mirror (air)	7.7 mm	10.9 mm
Rayleigh length (water)	2.64 mm	N/A
Rayleigh length (air)	2.0 mm	2.55 mm
Distance to zero delay	N/A	7.5 mm
Axial scan range	N/A	8.22 mm
Axial resolution after dispersion compensation (in air)	N/A	11 µm
Lateral resolution in focus (ω_0_)	N/A	28.6 µm
Output power at catheter tip	N/A	6 mW
Sweep frequency	N/A	200 kHz
Coherence length	N/A	>100 m [26]
Frames per second rate (after averaging and interlacing)	N/A	45–55 Hz

* During the measurements, it was noticed that the angle of the B-scan, and thereby the optimal MEMS frequency, varied slightly over time. ** Unavailable since the current catheter is not waterproof. MEMS, microelectromechanical systems; N/A, not available.

## Data Availability

The original contributions presented in this study are included in the article/Appendix A. Further inquiries can be directed to the corresponding author.

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
