# Peer review of "Ex Vivo Optical Coherence Tomography Analysis of Resected Human Bladder with a Forward-Looking Microelectromechanical Systems Mirror-Based Catheter"

_sensors, 2025, doi:10.3390/s25185794_

Round 1
Reviewer 1 Report
Comments and Suggestions for Authors
Dear Dr. Remmelink and co-authors,
I have reviewed your manuscript detailing the development of a forward-looking MEMS-OCT catheter for bladder imaging. Below are my comments and recommendations for improving the study’s rigor and clinical relevance.
First, there are several technical points that need clarification. The term "plural misfits" (Lines 53-54) is unclear in its current form - I suggest replacing it with "geometric mismatches" for better precision. In Table 1, the N/A designation for water-based focus measurements should be explained, particularly whether this was due to the prototype not being waterproof. The discrepancy between theoretical (30°) and experimental (varying) beam path angles warrants more discussion in the Results or Discussion section.
I noticed some inconsistencies that should be resolved. The MEMS mirror frequency is reported as both 550 Hz (Lines 117-119) and 562 Hz (Line 254) - please verify and correct this. Similarly, while the theoretical and experimental focus distances (11 mm vs 10.9 mm) align well, the minor discrepancy should be noted. For Figure 3A, the optimal angle/distance (7.84°, 7.46 mm) should be more clearly indicated to help readers interpret the data.
Regarding methodology, the sample handling procedure could be more clearly described. Please specify how the B-scan direction was marked for histopathology correlation (e.g., using sutures or ink markings). The "minor adjustments in angle" (Line 215) seems somewhat subjective - it would help to indicate whether these were performed manually or using software algorithms.
The imaging results raise some important questions. The inability to distinguish bladder layers (Lines 280-282) contrasts with previous studies; this should be discussed in the context of potential fixation artifacts and lack of physiological distension. For Figure 7I/J, more detail on how necrosis appears in OCT images would be valuable. The reported catheter breakdown (Line 264) suggests durability concerns that should be addressed regarding clinical translation.
Some additional analyses would strengthen the work. A comparative table analyzing your system against existing OCT technologies would help contextualize the performance tradeoffs, particularly regarding the 28.6 μm lateral resolution. The discussion of side-viewing catheters (Lines 296-318) could better emphasize why they're unsuitable for bladder imaging. For the B-scan angle discrepancy (Lines 335-346), considering MEMS damping or voltage limitations as potential causes would be worthwhile.
I recommend several improvements for future work:
-
Developing a waterproof version of the catheter to enable proper liquid environment testing
-
Conducting fresh tissue experiments to better assess true imaging capabilities
-
Implementing design modifications to address the fragility issues
-
Quantifying resolution differences between air and water imaging conditions
The manuscript would benefit from careful proofreading to address minor issues like the duplicate "correspondence" in Line 18 ." More importantly, explicitly linking ex vivo artifacts to the layer visibility challenges would strengthen the discussion section.
While the study shows promise, these revisions would significantly improve the manuscript's technical rigor and clinical relevance. I would be happy to review a revised version that addresses these points.
Thank you!
Reviewer 2 Report
Comments and Suggestions for Authors
Dear authors, I’ve read your article with great attention and pleasure. As a representative of the OCT community, I warmly welcome the use of technology for any non-ophtalmic purposes.
No doubt, your article is worthy of publication in Sensors journal, but I would like to make a number of comments to make this article better and more complete.
Thebladder you chose to demonstrate the capabilities of the forward looking probe you developed is certainly good, but it was used as an object of research quite densely and even in vivo mode. In my opinion, the main innovation in your article is, in fact, the forward looking probe, which is built somewhat differently than it was done before. It is the development of probe that makes your work to be in the scope of Sensors journal, but it is described very briefly, focusing more on the characteristics obtained than on the device itself. When reading the relevant section, I have a number of questions.
- What kind of MEMS element was used, who made it? At the same time, part. No or high-quality picture of the MEMS (if it is home-made) is required (you have specified the almost all the elements, including the IVS-1000 interferometer and DAQ card, but the most important element of the publication has been overlooked).
- What is the MEMS scanning algorithm? ("a square wave signal of 15 V" and other remarks in the text suggest resonant moving of the mirror, but am I right?) And why is “square wave”, when other shapes may be more compatible to control the mirror in this or any other regime?
- If the scanning is carried out in a resonant way, then how exactly is provided the synchronization of data acquisition to the MEMS mirror? And how stable is this synchronization (from one B-scan to another), what does the "approximately" sign mean when indicating the seemingly exact number of B-scans (rows 120, 122)?
- How does using two consecutive B-scans reduce the lateral step size? Offhand, several mechanisms can be assumed to achieve what is claimed, but they all have a limited scope, due to the ways in which scanning and synchronization of scanning and polling are performed. So I would like to ask for an accurate description in this place, to be sure, not guessing based on (my) own experience and the features of familiar(to me) devices.
- Which B-scans are being averaged in line 124 – "short" (364 A-scans each) or combined (728 A-scans)?
- How invariant is the velocity of the probing beam in the plane of the object, or how is the uneven movement of the mirror taken into account when restoring the OCT image?
- The 0-delay position should be painted on figs 3.b-c. Moreover, it would be better, if these figures were played some far from color map scale of fig.3.a. The proximity of the image to the numerical signatures causes misunderstandings in the perception of the image.
- Does the shape of OCT image of microscopy slide contain bevels? If yes, please point it in the text; it is important to understand what is depicted on fig.3.b-c (or put schematic drawing of experiment setup). Additionally, color drops marking slide edges are very difficult to be noticed (I had to enlarge the 3.b image almost to half of screen to see them. Please, chose another way to mark (if it is necessary) edges on the figure.
- In Table 1 and its preceding description, you indicate a fairly large difference between the model (Ansys Zemax) and the actual distance from the MEMS to the focus plane. This is quite typical when working with gradient optics, especially if short spacers are used, due to the exact manufacture of spacer is not always feasible in artisanal conditions. This is not a problem for the experiment, but it can be used to adjust the theoretical calculation. Perhaps this could correct the situation in which, with the expected same beam size on the green lens, in the calculated case, with a closer (30%) focusing, the Rayleigh length is 25% greater than in the experiment.
- The phenomenon of having some kind of "optimal" FOV and "0-delay distance" should be discussed. Probably, you are to use any target as Ronchi rulings (https://www.edmundoptics.com/f/precision-ronchi-rulings-on-opal-glass/13366/) to demonstrate the dependence of the distribution of lines over OCT image as a function of scanning range and 0-distance. From the proposed text, I could not understand the nature of this phenomenon, especially since “0-delay distance” is, in fact, a characteristic of the length of the reference arm of the interferometer, and it is not clear how it can affect the accuracy of lateral measurements.
- Specify the need to use the accuracy of determining the Rayleigh length (and some other values) up to 3-4 digits. In most cases, this is clearly excessive accuracy that makes no physical sense. Additionally, check the values listed in the text, figures and captions for consistency (as an example, the values for Rayleigh length 2.408 in line 234, 2.4 in FIG.6 and 2.5 in the caption to Fig. 6)
- Line 300 contains an inaccuracy regarding the commercialization of the catheters mentioned in [17]. As I know, these catheters, along with the OCT engine, were approved by the national regulator (in Russia, 2012) and are available for purchase (http://biomedtech.biz/en/products.php ) in several outer diameter versions: 1.8, 2.4 and 2.7 mm (unfortunately, detailed information is available only in Russian).
- When comparing the axial resolution of different OCT systems (lines 305-306), one should take in mind that this parameter is not related to the probe proposed in (and is the main character of the) publication, but is determined by the effective spectrum (sweeping) width of the OCT engine. Conversely, when comparing the lateral resolution, it is worth noting that the larger size of the spot in focus allows for a larger range of imaging, which is not critical for contact probes (in [17] (and [29], since it is, in fact, the same probe), the probe is fundamentally contact), given the shallow depth of visualization of most biological tissues, but with non-contact sensing it becomes a significant advantage.
- The fact that you do not "see" the structural elements of the tissue (line 343) may be caused by many factors, not only the ex vivo observation format, the evenly stretching of the bladder in its natural state, and the somehow modified structure of your samples. To a large extent, this is facilitated by a sharper tissue boundary (in your case, this is the tissue/air boundary, whereas in vivo it is tissue/water), and an attempt to correctly display this boundary leads to an effective reduction in the dynamic range of the visualized image in the area of the tissue itself. This is complemented by the incorrect operation of the image reconstruction algorithm (as seen in FIG. 7.g-k). In individual images (FIG.7.a-e), the target image fragments look much better, but their small size (both by itself and in comparison with the large uninformative part of the image) does not allow you to see the structural elements. It may be worth working on increasing the dynamic range, as well as on suppressing artifacts caused by long coherence of used probing light and heterodyne effects while spectrum sampling. In addition, it would not be superfluous to note that determining the longitudinal resolution based on the width of the restored hardware function is acceptable, but somewhat different from determining the ability to distinguish between two objects located at a minimum distance from each other. From the point of view of the essence of the presented work and the journal to which it is directed, it would be advisable to conduct a more objective analysis of the resolution of the device using special phantoms or more easily accessible in vivo (and familiar to the technical specialist) tissues (thumb skin with sweat gland ducts, for example).
Reviewer 3 Report
Comments and Suggestions for Authors
The peer-reviewed scientific paper makes a very good impression. The authors conducted a balanced research, including: I) design and assembly of a swept source optical coherence tomography setup with a replaceable sample arm; II) design and manufacture of an endoscopic forward-view probe; III) ensuring hardware and software compatibility; IV) testing of the obtained system; V) optimization of its operating modes; VI) biomedical applications their system. The topic of the research is very relevant due to the presence of a partially unrealized opportunity to replace targeted biopsy with its non-invasive analogue in the form of optical biopsy with an endoscopic probe of the optical coherence tomography system.
The paper is recommended for publication with minor revisions. In particular, the reviewer draws the authors' attention to the fact that:
I) The sentence «OCT utilizes infrared light to create high-resolution depth resolved cross-sectional images of tissue structures by measuring the backscattered light» (line 61) is somewhat incorrect. The source of useful information in optical coherence tomography is not only backscattered photons, but also back-reflected photons and, in some cases, photons with short trajectories (several interactions). In addition, systems for optical coherence tomography are known with a wavelength of probing radiation falling in the red region of visible radiation. This part of the paper needs to be clarified.
II) The reasons for developing a system for swept source optical coherence tomography are not obvious. An optical coherence tomography system with an endoscopic probe for urological purposes could well have been made, for example, based on a superluminescent diode. It would be advisable to explain the reasons for the author's choice.
III) It would not be superfluous to mention the general structure of the scanning head of the endoscopic probe. For example, how the issues of guiding when the probe moves, choosing of region of interest and visualisation corresponding to it the scanning line, washing the lenses from mucus and other biological fluids and so on are resolved. Most likely, the author's version of the endoscopic probe is compatible with multichannel probes for urology and the above issues have already been resolved there. Nevertheless, this should have been stated explicitly.
IV) The authors phrases about theoretical spatial resolution are a bit ambiguous. Strictly speaking, the theoretical limit of spatial resolution is half the wavelength of the probing radiation. In the context of paper, authors are talking about the theoretical characteristics of the system being developed and the characteristics that were obtained in reality (and they are quite impressive).
In the opinion of the reviewer, taking into account the above comments will make the scientific paper more understandable to readers. The reviewer also considers a second review unnecessary due to the initially high level of the work presented by the authors.
Round 2
Reviewer 2 Report
Comments and Suggestions for Authors
Dear authors, I would like to thank you for the work you have done to enhance the material of your article, but I am confused that you are still avoiding the issue of describing the MEMS element used. After all, the journal is called “Sensors”, and the sensor is the subject of the publication. In your case, the key element of the sensor is classified, the illustrative material is limited to a 3D model of the sensor and a rather poor photograph of the tissue sample and the external mechanical strapping of the sensor in the experiment.
In your response to a comment about the MEMS mirror used, you pointed out that "The MEMS mirror is a proprietary design of Scinvivo," but in my opinion, this information is not enough, since it is precisely the fact that you found a good MEMS mirror and managed to build a probe head on it that is the main aspect of the novelty of your work. At least indicate the patent number and manufacturer – and not in the answers to the reviewer, but directly in the text of the article. In addition to the above, it should be added that the MEMS mirror has positioning control systems (I draw this conclusion from your answer) - this, again, is important from the point of view of describing the sensor and the mechanism by which you get one with the best resolution from two consecutive B-scans.
Row 81. I may be finding fault, but traditionally, capital letters are used in writing "A-scan" and "B-scan" (in line 122 and beyond, you use the usual spelling).
Row 380 The difference between the simulated and experimental focus distance of 0.1 mm with a Rayleigh length of about 3 mm is so insignificant that it should hardly be mentioned in the work. However, the question arises, how does this relate to the value of 7.7 mm in table 1 and line 236?
Author Response
In your response to a comment about the MEMS mirror used, you pointed out that "The MEMS mirror is a proprietary design of Scinvivo," but in my opinion, this information is not enough, since it is precisely the fact that you found a good MEMS mirror and managed to build a probe head on it that is the main aspect of the novelty of your work. At least indicate the patent number and manufacturer – and not in the answers to the reviewer, but directly in the text of the article.
Answer: We thank the reviewer for his comment, we added the patent number and manufacturer directly in the text, line 101-102: ‘An optomechanical catheter (Scinvivo BV, Eindhoven, the Netherlands, patent number: NL2020564B1), was designed for eventually in-vivo imaging, housing both the focussing optics and the MEMS mirror.’
In addition to the above, it should be added that the MEMS mirror has positioning control systems (I draw this conclusion from your answer) - this, again, is important from the point of view of describing the sensor and the mechanism by which you get one with the best resolution from two consecutive B-scans.
Answer: To clear this up, we added the following sentence in line 123-124: ‘The ASIC driver further comprises a feedback input to receive a feedback signal indicative for a rotational state of the rotor vs the stator of the MEMS mirror.‘
Row 81. I may be finding fault, but traditionally, capital letters are used in writing "A-scan" and "B-scan" (in line 122 and beyond, you use the usual spelling).
We agree with the reviewer and changed the wording into A-scan and B-scan
Row 380 The difference between the simulated and experimental focus distance of 0.1 mm witha Rayleigh length of about 3 mm is so insignificant that it should hardly be mentioned in the work.However, the question arises, how does this relate to the value of 7.7 mm in table 1 and line236?
Answer: We thank the reviewer for this observation. We went back to the Zemax simulation and redid the Rayleigh length calculation again. This resulted in some differences mainly because we used the wrong input for the fibre NA. We therefor changed the Rayleigh length values in the table and text accordingly in lines 238 – 247.
As to your question on how this relates to the value of 7.7 mm obtained from the simulation? We observed an increase in the distance of the focal position relative to the MEMS mirror for the experimental measurement compared to the Zemax predictions.
We therefor added the following text to the discussion, line 286-391: ‘This can be explained by small differences in the real optical system compared to the simulated one. Through zemax simulations, it was found that a 50 micron increase in the length of the GRIN component could explain the observed shift. Such a shift would also match the observed increase in spot size and corresponding increase in Rayleigh length. It should be noted, however, that these deviations of the experimental beam profile do not compromise imaging quality.’